# Clinical Theragnostic Potential of Diverse miRNA Expressions in Prostate Cancer: A Systematic Review and Meta-Analysis

**DOI:** 10.3390/cancers12051199

**Published:** 2020-05-09

**Authors:** Rama Jayaraj, Greg Raymond, Sunil Krishnan, Katherine S. Tzou, Siddhartha Baxi, M. Ravishankar Ram, Suresh Kumar Govind, Harish C. Chandramoorthy, Faisal N. Abu-Khzam, Peter Shaw

**Affiliations:** 1Northern Territory Medical Program (NTMP), College of Medicine and Public Health, Flinders University, CDU Campus, Ellengowan Drive, Darwin, Northern Territory 0909, Australia; 2Department of Artificial Intelligence, Nanjing University of Information Science and Technology (NUIST), Nanjing, Jiangsu 210044, China; faisal.abukhzam@lau.edu.lb (F.N.A.-K.); 100001@nuist.edu.cn (P.S.); 3Flinders University Northern Territory Medical Program, CDU Campus, Ellengowan Drive, Darwin, Northern Territory 0909, Australia; greg.raymond@flinders.edu.au; 4Department of Radiation Oncology, Mayo Clinic Florida, 4500 San Pablo Road S., Jacksonville, FL 32224, USA; krishnan.sunil@mayo.edu (S.K.); tzou.katherine@mayo.edu (K.S.T.); 5GenesisCare Gold Coast Radiation Oncologist, John Flynn Hospital, 42 Inland Drive, Tugun, QLD 4224, Australia; Siddhartha.baxi@genesiscare.com; 6Department of Genetics and Molecular Biology, Institute of Biological Sciences, Faculty of Science, University of Malaya, Kuala Lumpur 50603, Malaysia; mravishankarram@um.edu.my; 7Faculty of Medicine, University of Malaya, Kuala Lumpur 50603, Malaysia; suresh@um.edu.my; 8Stem Cells and Regenerative Medicine Unit, Department of Microbiology and Clinical Parasitology, College of Medicine, King Khalid University, Abha 61421, Saudi Arabia; hshkonda@kku.edu.sa; 9Department of Computer Science and Mathematics, Lebanese American University, Beirut 00000, Lebanon; 10Menzies School of Health Research, Darwin 0810, Australia

**Keywords:** miRNA, theragnostic potential, chemoresistance, systematic review, meta-analysis

## Abstract

*Background:* Prostate cancer (PrC) is the second-most frequent cancer in men, its incidence is emerging globally and is the fifth leading cause of death worldwide. While diagnosis and prognosis of PrC have been studied well, the associated therapeutic biomarkers have not yet been investigated comprehensively. This systematic review and meta-analysis aim to evaluate the theragnostic effects of microRNA expressions on chemoresistance in prostate cancer and to analyse the utility of miRNAs as clinical theragnostic biomarkers. *Methods:* A systematic literature search for studies reporting miRNA expressions and their role in chemoresistance in PrC published until 2018 was collected from bibliographic databases. The evaluation of data was performed as per PRISMA guidelines for systematic review and meta-analysis. Meta-analysis was performed using a random-effects model using Comprehensive Meta-Analysis (CMA) software. Heterogeneity between studies was analysed using Cochran’s Q test, I^2^ and the Tau statistic. Quality assessment of the studies was performed using the Newcastle–Ottawa Scale (NOS) for the methodological assessment of cohort studies. Publication bias was assessed using Egger’s bias indicator test, Orwin and classic fail-safe N test, Begg and Mazumdar rank collection test, and Duval and Tweedie’s trim and fill methods. *Findings:* Out of 2909 studies retrieved, 79 studies were shortlisted and reviewed. A total of 17 studies met our eligibility criteria, from which 779 PrC patients and 17 chemotherapy drugs were examined, including docetaxel and paclitaxel. The majority of the drug regulatory genes reported were involved in cell survival, angiogenesis and cell proliferation pathways. We studied 42 miRNAs across all studies, out of which two miRNAs were found to be influencing chemosensitivity, while 21 were involved in chemoresistance. However, the remaining 19 miRNAs did not appear to have any theragnostic effects. Besides, the prognostic impact of the miRNAs was evaluated and had a pooled HR value of 1.960 with 95% CI (1.377–2.791). *Interpretation:* The observation of the current study depicts the significance of miRNA expression as a theragnostic biomarker in medical oncology. This review suggests the involvement of specific miRNAs as predictors of chemoresistance and sensitivity in PrC. Hence, the current systematic review and meta-analysis provide insight on the use of miRNA as PrC biomarkers, which can be harnessed as molecular candidates for therapeutic targeting.

## 1. Introduction

Prostate cancer (PrC) is the fourth-most common cancer across genders and in males is the second-most common. Based on the Globocan 2018 statistics, the total number of new cases was 1,276,106 (7.1%) with a death rate of 4.5% (432,242 patients). Australia and New Zealand had the highest age-standardized incidence as well as mortality rates with 86.4 and 10.2 per 100,000 people respectively [1]. Several studies highlight the incidence and mortality rate of PrC and an increased prevalence of PrC worldwide [2].

Some commonly preferred treatment methods for PrC are surgery [3], chemotherapy [4], radiotherapy [5] and hormone therapy [6]. Both neoadjuvant and adjuvant chemotherapies and a variety of chemotherapeutic drugs, such as docetaxel [7], paclitaxel [8], cisplatin [9], azacytidine [10], cyclopamine [11], estramustine, trichostatin A and thapsigargin [12] are reported in various studies, while paclitaxel is the only drug approved by the FDA for the treatment of prostate cancer [13]. According to a recent report published in 2017, androgen deprivation therapy (ADT) has been used to treat advanced stages of prostate cancer [14]. Taxane is a standard treatment for androgen-independent prostate cancer (AIPC) [15]. One of the foremost reasons for the failure of the treatments is the development of chemoresistance in PrC patients during long-term chemotherapy. There is emerging evidence substantiating the development of resistance to several forms of therapy in PrC [16,17,18,19]. Multiple drug resistance (MDR) in PrC may be due to various mechanisms, including ABC transporters, stress-mediated conditions (glucose starvation and hypoxia), proteasome inhibitions, glyoxalase I and Akt. There have been minimal reviews elaborating on mechanisms relating to chemotherapy drug resistance [20,21,22]. Initial responders to docetaxel treatment are found to develop resistance over the treatment period [21,22].

A study by Cochrane et al. 2009 shows that miRNAs are involved in chemoresistance in many cancers [23]. The development of chemoresistance has been associated with changes in miRNA expression during prostate cancer progression [24]. Chemoresistance to PTX in prostate cancer cells was due to the altered expression of miRNAs, and treatment with antimitotic drugs with small molecules reverted the chemoresistance by targeting the miRNAs involved in chemoresistance [11]. Combination drugs using docetaxel, cyclopamine and gefitinib were shown to induce increased anti-proliferative and anti-apoptotic effects on metastatic prostate cells relative to individual medications [25].

Chemosensitivity is the measure of the susceptibility of a particular drug towards the tumor. A previous investigation of 56 cancers and seven normal tissues revealed that 39 miRNAs were upregulated and six miRNAs were downregulated in PrC [26]. Both downregulation of 37 miRNAs and upregulation of 14 miRNAs were observed in other PrC studies in 6 cell lines, 9 xenografts, 13 cancer tissues and 4 benign tissues [27]. miRNA-145 levels were analysed in laser capture microdissected (LCM), PrC tissues and 47 cancer cell lines and were found to be downregulated, with a significant correlation being observed between miRNA-145 and p53 via DNA methylation [28].

Another study in PrC samples evidenced that miRNA-101 regulates the enhancer of zeste homologue 2 (EZH2) [29]. There is emerging evidence establishing the role of miRNA-143 in sensitising the PrC cells to docetaxel treatment through suppression of KRAS [30]. Another study by Yu. J and colleagues shows the effects of miR-200b in suppressing cell proliferation and enhancing the chemosensitivity of PrC through regulation of Bmi-1 [31]. As more studies emerge involving miRNAs, it is has been recently observed that the basal cell layer from which the PrC emulates possesses progenitors or adult stem cells [32]. Though the presence of the stem cells themselves is debated in the prostate, the origin of miRNA and the presence of stem cells opens the opportunity for other research approaches in biomarker studies yet to be fully elucidated.

The challenge in choosing a therapy for prostate cancer is the inconsistent levels of PSA in the serum, due to which alternative markers are essential for firm establishment of the correct treatment [33]. miRNAs are reliable predictors, and numerous studies have developed and validated the use of miRNAs as a potential indicator of prostate cancer [34].

Thus, there is emerging evidence establishing the importance of miRNA expression in the development of drug resistance in PrC [24]. However, the effects of miRNA expression on chemoresistance and sensitivity have not been comprehensively studied in the form of a systematic review and meta-analysis. Therefore, this systematic review and meta-analysis aimed to investigate the regulatory role of miRNA in prostate cellular processes and their involvements in the development of chemoresistance in PrC.

## 2. Methods

### 2.1. Search Strategy and Study Selection

PubMed and Science Direct were used to identify relevant literature published until December 2018. The search strategy aimed at collecting articles relating to miRNA, chemoresistance and prostate cancer. The search terms used were “miRNA or microRNA” AND “chemoresistance” AND “prostate cancer”. Only studies that were published in English or had official English translations were included in this study. Manual screening of the reference lists of eligible studies was performed to identify additional relevant articles. Full-text articles were scrutinised after initial screening by titles and abstracts. The final selection was based on predefined inclusion and exclusion criteria. The corresponding authors were contacted for collecting pertinent data if it was found missing in the full texts of the articles. Duplicates were removed, and the study was excluded if it fell within the exclusion criteria (Appendix A).

### 2.2. Selection Criteria

All retrieved articles were evaluated by the reviewers (R.J., S.B., M.R.R.) for selection based on the following criteria:

### 2.3. Inclusion Criteria

Our primary inclusion criteria were the studies analysing the theragnostic effect of miRNA expressions in both prostate cancer patients and cell lines.

Inclusion criteria were:Studies reporting chemoresistance in PrC.Studies reporting a miRNA profiling platform.Studies on drug regulatory genes or pathways involved in chemoresistance or sensitivity.Studies reporting miRNA expressions analysis using in-vitro assays on chemoresistance.

### 2.4. Exclusion Criteria

Studies published in languages other than English.Letters to the editor, case studies, review articles and studies performed only in PrC patients or in vitro.Studies using PrC patients’ information from GenBank datasets.

Disagreements between reviewers were resolved by discussion and consultation with the corresponding author and fifth reviewer.

### 2.5. Data Extraction and Analysis

Studies that complied with the selection criteria were used for extracting study data. Data from the included studies were obtained by S.B., M.R.R., and P.S. and cross-checked by R.J. Corresponding authors of selected articles were contacted for further clarifications and Appendix A if required. The following list of data items was extracted from the full-text articles and Appendix A and recorded in an MS Excel (master sheet) data extraction form, designed based on PRISMA guidelines [35];
First author and year of publicationCountryPatients’ originEthnicityNumber of samplesCell linesResistant cell lines to chemotherapymiRNA(s) involvedmiRNA profiling platformDrug informationMolecular pathways or gene associated

### 2.6. Quality Assessment

The Dutch Cochrane Centre’s Meta-Analysis of Observational Studies in Epidemiology (MOOSE) guidelines [36] were used to assess the quality of the included studies as adopted in previous studies [37,38]. All the criteria must have been mentioned in the main text, Appendix A or later provided by the corresponding authors to qualify for systematic review. In addition to that, the quality assessment of the included studies was performed using the Newcastle–Ottawa Scale (NOS) for the methodological assessment of cohort studies.

### 2.7. Publication Bias

Publication bias is an integral part of a systematic review and meta-analysis [39,40]. The inverted funnel plot depicts the publication bias. Publication bias was quantified using Egger’s bias indicator test [41], Orwin [42] and classic fail-safe N test, Begg and Mazumdar rank collection test, and Duval and Tweedie’s trim and fill calculation [43].

### 2.8. Meta-Analysis

Comprehensive Meta-Analysis (CMA) software 3.0 was used to analyse the pooled hazard ratio (HR) with 95% confidence interval (CI). In case of a lack of between-study heterogeneity, fixed model effects were used, and if not, random model effects were employed [44,45,46,47]. Possible influences such as number of patients, year of publication, study period, study locations, type of studies and diagnostic procedures were investigated for heterogeneity using Cochran’s Q test and Higgins I-squared statistic [48]. The statistic tau squared was used to study the variance between the studies and where it incorporates a threshold effect [39]. A Q test was used to differentiate between the observed effect and fixed effect model by summing up and squaring their differences [49].

## 3. Results

### 3.1. Search Strategy and Study Selection

Initial search identified 2909 relevant studies from PubMed (*n* = 179) and ScienceDirect (*n* = 2730). After a thorough screening, 2830 articles were removed for being duplicates, irrelevant, reviews, case studies and letters to the editor. Screening based on inclusion criteria was used to narrow down to 79 potentially eligible studies, and further screening based on exclusion criteria resulted in 17 articles. Figure 1 depicts a flowchart describing our selection process.

We identified 17 studies involving 779 PrC patients eligible for the systematic review. Table 1 shows the main characteristics of the included studies for the systematic review and meta-analysis. The included studies were conducted between 2005 and 2015. The majority of studies were performed in China (*n* = 11) followed by USA (*n* = 4) and one each from Australia and Austria. The two most preferred chemotherapy agents were docetaxel and paclitaxel.

### 3.2. In-Vitro Assays

The common in-vitro assay types and related number of studies are represented in Figure 2A, and the cell line types in Figure 2B. A total of 14 different cell lines were used in the 17 included studies, of which PC3 and DU145 were the most commonly used (in 12 and 8 studies, respectively). The highest number of cell lines used in a single study was 5 [12]. The in-vitro and in-vivo assay information collected from the included studies showed the use of -RT-PCR, luciferase assay, western blotting, chemotherapy sensitivity assay, apoptotic assay, cell viability assay, cell migration, cell proliferation, immunohistochemistry, chromatin immuno-precipitation (ChIP) assay, clonogenic assay, spheroid assay and caspase assay to determine miRNA expression and activity.

### 3.3. Association between miRNA Expression and Chemoresistance

A total of 42 miRNAs were studied as observed in the pooled study data of this systematic review. Of these, 34 miRNAs were observed to be downregulated, and six were upregulated. A total of 19 downregulated miRNAs (miR-1, 29b, 30d, 31, 34a, 125a-3p, 143, 144, 181a, 182, 199a, 199a-5p, 200b, 200c, 203, 204 and 205) were correlated with an increase in chemoresistance and two downregulated miRNAs (miR-34a and 212) appeared to be associated with increased chemosensitivity. It was also seen that three upregulated miRNAs (miR-10b, 17 and 155) correlated with increased chemoresistance. It was found that 15 downregulated miRNAs (miR-20a, 20b, 21, 22, 25, 132, 146a, 200a, 200b, 200c, 375, 429, 495, 590-5p and 4319) and three upregulated miRNAs (miR-132, 222 and 301b) had no effect on chemoresistance or sensitivity. High expression was observed in miRNA-10b, 17, 132, 155, 222, 301b and low expression of miRNA-1, 20a, 20b, 21, 22, 25, 29b, 30d, 31, 34a, 125a-3p, 132, 143, 144, 146a, 181a, 199a, 199a-5p, 200a, 200b, 200c, 203, 204, 205, 212, 222, 375, 429, 495 and 590-5p was observed. Of the 17 chemotherapy drugs studied, docetaxel is the most commonly used. Overall, 21 microRNAs were linked to the development of chemoresistance, and two microRNAs were linked to chemosensitivity.

### 3.4. Chemotherapy and PrC Patients

A total of 17 drugs were used as chemotherapy in the pooled studies; docetaxel (332 patients), paclitaxel (77 patients), dihydrotestosterone (DHT) (40 patients), AR inhibitor (MDV3100) (40 patients), cisplatin (40 patients), bicalutamide (30 patients), azacytidine (25 patients), DZNep (8 patients), tyrosine kinase inhibitors (25 patients), cyclopamine (3 patients), AR expression (30 patients), topotecan (25 patients), doxorubicin (25 patients), trichostatin A (7 patients), estramustine (40 patients), androgen-deprivation therapy (25 patients) and thapsigargin (7 patients).

### 3.5. Drug Regulatory Pathways for miRNA-Mediated Chemosensitivity and Chemoresistance

The miRNA-mediated chemoresistance pathways are represented in Figure 3. Of the 17 articles, 14 different pathways and their associated genes were elaborated upon in the individual studies. Seven pathways were described as leading to cell survival and six pathways were assessed to be involved in cell differentiation and proliferation, while one was linked to angiogenesis.

### 3.6. Association between miRNAs and Drug Regulatory Pathways of Chemoresistance

miRNA-34a, 200b, 200c and 205 were studied in two studies each. It was observed that miRNA-34a was downregulated on treatment in two studies (azacytidine, topotecan and doxorubicin; paclitaxel and cyclopamine) and was found to influence the AMPK/mTOR and Hedgehog signalling pathways thereby inducing chemosensitivity and resistance, respectively. PrC treatment with docetaxel was associated with downregulation of miRNA-200b through activation of the BMI-1 gene, contributing to chemoresistance. In another report, miRNA-200c was found to be downregulated during the treatment of docetaxel, paclitaxel and cyclopamine, which in turn triggered the E-cadherin and Hedgehog pathway leading to chemoresistance. Finally, miRNA-205 downregulation was observed on the treatment of docetaxel and DZNep resulting in chemoresistance mediated through EZH2 and E-cadherin gene (Table 2 and Table 3).

The association between miRNA expressions and patient survival was analysed using HR and 95% CI values through meta-analysis. High expression of miR-132 (HR = 1.9; 95% CI = 1.1–3.2) and low expression of miR-20a (HR = 1.8; 95% CI = 1–3.3), 21 (HR = 2.3; 95% CI = 1.3–4), 200a (HR = 2.8; 95% CI = 1.6–5), 200b (HR = 3.2; 95% CI = 1.7–6), 200c (HR = 2.3; 95% CI = 1.3–4.1), 375 (HR = 2; 95% CI = 1.1–3.6), 429 (HR = 3.5; 95% CI = 1.8–6.9) and 590-5p (HR = 0.5; 95% CI = 0.3–0.8) was observed. Out of 17 articles, only one article reported the HR and 95% CI value. The study that reported the HR values for nine miRNAs was used for this meta-analysis. A pooled HR value of 1.960 with a 95% CI (1.377–2.791; *p*-value: 0.000) was obtained (Figure 4).

### 3.7. Meta-Analysis

#### 3.7.1. Publication Bias

The funnel plot represented in Figure 5 is slightly asymmetrical across the studies, which indicates the presence of publication bias. The funnel plots from Figure 5 and Figure 6 have been constructed using the software ‘Comprehensive Meta-Analysis Software’ (version 3.3.070, USA). Each funnel plot was constructed alongside the forest plot using HR values and 95% CI for each cohort. Each point in the funnel plot represents an individual cohort. If these points were symmetrically present across the regression line, it would indicate the lack of any publication bias in the conducted meta-analysis. Whereas, in the current study, the points in the funnel plot are not symmetric, indicating the existence of bias.

#### 3.7.2. Classic Fail-Safe N

This meta-analysis incorporates data from nine studies, which yield a z-value of 6.82789 and a corresponding two-tailed *p*-value of 0.00001. The fail-safe N is 101. This means that we would need to locate and include 101 ‘null’ studies for the combined two-tailed *p*-value to exceed 0.050. This could alternatively be expressed that there would be 11.2 missing studies for every observed study for the effect to be nullified.

#### 3.7.3. Orwin Fail-Safe N

Here, the hazard ratio in observed studies is 1.96, which did not fall between the mean hazard ratio in the missing studies, so we could not calculate the Orwin fail-safe N.

#### 3.7.4. Begg and Mazumdar Rank Correlation Test

In this case, Kendall’s tau b (corrected for ties, if any) is 0.41667, with a one-tailed *p*-value (recommended) of 0.05893 or a two-tailed *p*-value of 0.11785 (based on continuity-corrected normal approximation).

#### 3.7.5. Egger’s Test of the Intercept

In this case, the intercept (B0) is 18.97814, 95% confidence interval (6.27343, 31.68286), with *t* = 3.53225, df = 7. The one-tailed *p*-value (recommended) is 0.00478, and the two-tailed *p*-value is 0.00957.

#### 3.7.6. Duval and Tweedie’s Trim and Fill

This method suggests that three studies are missing (Figure 6). Under the fixed-effect model, the point estimate and 95% confidence interval for the combined studies are 1.86926 (1.54497, 2.26162). Using trim and fill, the imputed point estimate is 1.52342 (1.28781, 1.80214). Under the random-effects model, the point estimate and 95% confidence interval for the combined studies are 1.99148 (1.31951, 3.00566). Using trim and fill, the imputed point estimate is 1.56458 (1.06512, 2.29825).

Publication bias analysis of the included studies was conducted to study the effect of publication bias on the results of this study. Figure 6 shows the funnel plot results with imputed studies. The symmetric nature of the plot denotes the presence of no bias. From the funnel plot, it is obvious that the smaller included studies place towards the bottom of the funnel plot, and the more extensive studies look towards the top of the graph, with clustering near the mean effect size. Large studies appear outside the funnel and tend to cluster on one side of the funnel plot. Smaller studies appear toward the top of the graph, since there is more sampling variation in effect size estimates in the smaller studies, which will be dispersed across a range of values.

## 4. Discussion

The drug resistance mechanisms in PrC are being extensively studied, as indicated by the emerging studies [20,59,60,61,62]. The purpose of this systematic review is to investigate and evaluate the miRNA biomarkers as potential predictors in chemoresistance/sensitivity in PrC, and their association with different drug-regulatory genetic pathways.

The previous meta-analysis on prostate cancer was carried out deciphering the role of miRNAs targeting the androgen receptor [63]. miRNA-21 is downregulated in our study, similar to in another report which demonstrates that the reduction in expression could inhibit the cancer growth, while its overexpression in androgen-independent prostate cancers makes them resistant to androgen ablation [64,65].

The contradicting results of miRNA-375 expression in one of our studies show it is downregulated while another study reported upregulation of this miRNA in serum samples, indicating their role in the development of AIPCs [66]. miRNA-34a, 146a, 205 are downregulated similar to the results stated in the Li and Mahato (2014) review [67]. Synthetic miRNAs or genetic precursors can be used to restore the levels of tumour-suppressive miRNAs, which is known as miRNA replacement therapy [68].

miRNA-10b is a diagnostic marker for glioma [69], breast [70], oral [71] and pancreatic cancer [72]. miRNA-20a is used as a potential diagnostic biomarker for cervical cancer [73], nasopharyngeal cancer [74] and pancreatic cancer [75]. miRNA-200c is used as a novel prognostic marker in colorectal [76], gastric [77], ovarian [78] and lung cancer [79]. miRNA-21 expression has proven to be useful as a therapeutic biomarker in pancreatic [80], breast [81] and non-small lung cancer [82]. These miRNAs have been studied concerning their drugs and pathways and are included in this review.

EZH2 plays a key role in cancer progression [83], and its function as a key regulator is studied in several cancers, such as breast [84], prostate [29] and nasopharyngeal carcinomas [85]. Hedgehog functions as a mediator of tumorigenesis, and its signalling is essential in the regulation of cancer and tumorigenicity [86]. Of the numerous existing RAS proteins, KRAS has a crucial role in proto-oncogene activity, and its mutation status has been highly explored for therapeutic response in cancer research [87]. E-cadherin is an invasion- and tumour-suppressor protein [88] and plays a role in the transition of adenoma to carcinoma, and its repressed expression is a poor prognostic indicator in cancer [89,90]. ZEB1 and its two proteins are studied widely as they promote the epithelial–mesenchymal transition in cancer and are direct repressors of E-cadherin [91]. TCF7 activates transcription through the Wnt/β-catenin pathway and is observed to be involved in the growth and metastasis of cancer [92,93].

PC3 and DU145 cell lines are being used for evaluating transcriptional activity [94], in cancer studies [95,96,97] and for evaluating genetic activities [98,99], which is examined in most of the authors’ included studies. The results of this meta-analysis show a significant pooled effect when correlating drug-resistance-related miRNAs to patient survival, thereby reinforcing the possibility of use of miRNA as prognostic markers. However, we also have to take into consideration that only one study reported the HR and 95% confidence interval.

### 4.1. Strengths of Our Study

This systematic review and meta-analysis follow appropriate best practice research and statistical guidelines, and the following points accentuate this approach. Initially, the studies chosen were selected from a global arena, where there is a diversity in the selection of patients and their outcomes; secondly our selection criteria (which follow the PRISMA guidelines) help us to engage and have access to an extensive global literature database. An analysis of the publication bias was minimised to minimise the heterogeneity between the studies selected in our meta-analysis.

This comparison will help future researchers to evaluate and publish articles from separate patients, which will collectively add to the knowledge base around miRNA prognosis in PrC patients. From this point of view, this study brings out more value for future researchers and technicians regarding predicting miRNA as a valuable prognostic biomarker. The use of subgroup analysis also took into consideration demographic characteristics and repeated miRNAs to provide a better understanding of the survival outcomes of PrC patients. Additionally, the authors believe that this is the foremost systematic review and meta-analysis study on the prognostic utility of miRNAs in PrC patients.

### 4.2. Limitations of Our Study

The majority of the included studies in our meta-analysis were on patients from Asian and Caucasian backgrounds, which might limit the applicability of the results. From the prognostic point of view of these specific miRNAs, only one study reported the HR and 95% CI value, which is a major limitation, as the rest of the values were estimated from KM curves. Finally, potential biases and confounders cannot be avoided completely, since all studies included had observational designs. Our hallmarks of the PrC provide an overview of the miRNAs involved in drug resistance. The identified miRNAs can be used as diagnostic, prognostic and therapeutic biomarkers in PrC.

### 4.3. Future Work

It is expected that analysing the graph based on the correlation between pairs of miRNAs using cluster editing will reveal other drug regulatory pathways that are not detected yet. This is because clusters (cliques) of mutually pairwise correlated miRNAs represent a strong indication of potential chemoresistance pathways. In fact, clustering can be used as a verification phase or to possibly disclose other pathways that are not detected yet.

## 5. Conclusions

There is extensive anticipation regarding the use of miRNAs as a possible predictor of chemoresistance in prostate cancer as the available markers are unreliable. There are a significant number of published studies on the evaluation of miRNA regulation in prostate cancer. Our systematic review and meta-analyses from recent clinical evidence demonstrate that miRNAs facilitate drug resistance and sensitivity in prostate cancer patients. The hallmarks of specific miRNAs in prostate cancer as highlighted by our study might provide insights into the regulation of miRNAs and the genes associated with the process of drug resistance, thereby helping future clinicians and researchers in designing efficient clinical trials and in-vitro studies. We have provided a list of miRNAs and their respective pathway targets for routine therapeutic purposes. Further research is required to highlight which specific miRNAs may be intricately involved in chemoresistance and sensitivity.

## Figures and Tables

**Figure 1 cancers-12-01199-f001:**
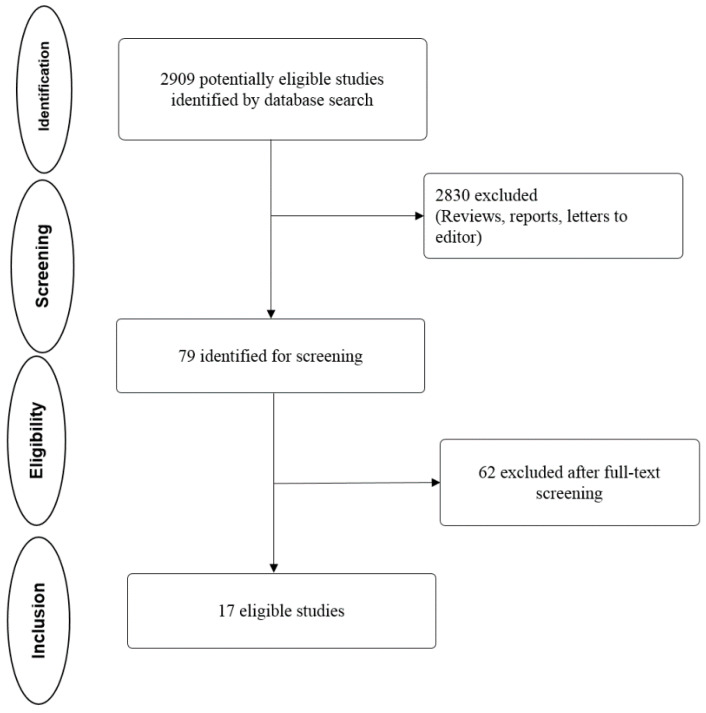
Flowchart of the literature study process and selection.

**Figure 2 cancers-12-01199-f002:**
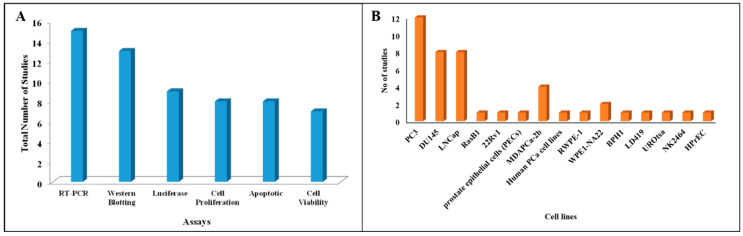
(**A**) A number of in-vitro assays from the included studies, exploring the association between miRNA expressions and chemoresistance; (**B**) the number of studies involving cell line variants.

**Figure 3 cancers-12-01199-f003:**
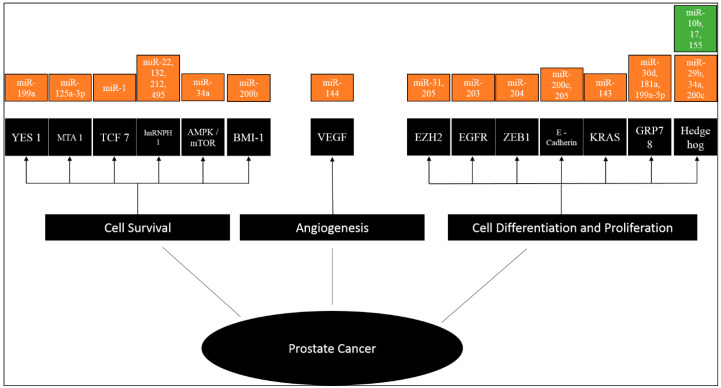
Hallmarks of specific miRNAs in prostate cancer. YES1, MTA1, TCF7, hnRNPH1, AMPK/mTOR, BMI-1, VEGF, EZH2, EGFR, ZEB1, E-Cadherin, KRAS, GRP78, Hedgehog. Each hallmark shows some examples of miRNAs that influence particular cellular functions in PrC; some microRNAs influence more than one hallmark indicating multiple pathways regulated by them. Green—upregulated; Orange—downregulated.

**Figure 4 cancers-12-01199-f004:**
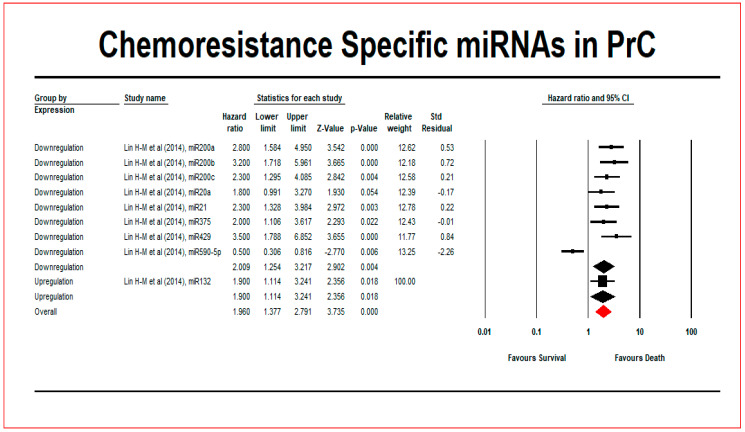
Title: meta-analysis of chemoresistance-specific miRNA in PrC patients. Figure 4 legend: The black diamond from Figure 4 graphically represents the pooled effect estimate of survival for PrC patients randomly assigned to evaluate the chemoresistance specific miRNA. The vertical line and red box indicate the effect size of miRNA expression in the included studies with a 95% CI. A HR of 1 suggests no difference in risk in PrC patients.

**Figure 5 cancers-12-01199-f005:**
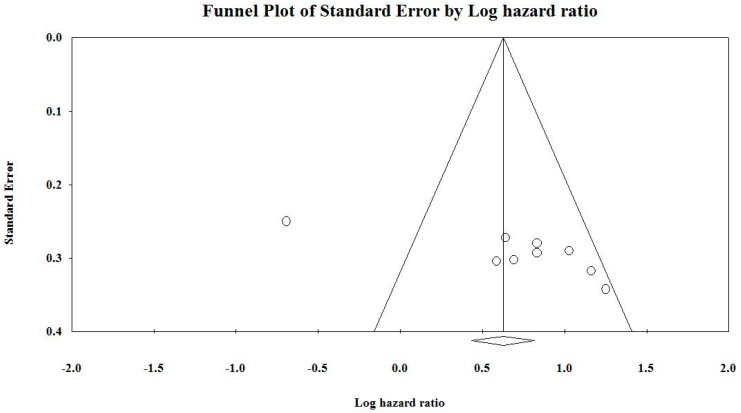
Funnel plot of studies included in the meta-analysis. Each study is represented as clear circles, and the studies outside the line represent the asymmetry.

**Figure 6 cancers-12-01199-f006:**
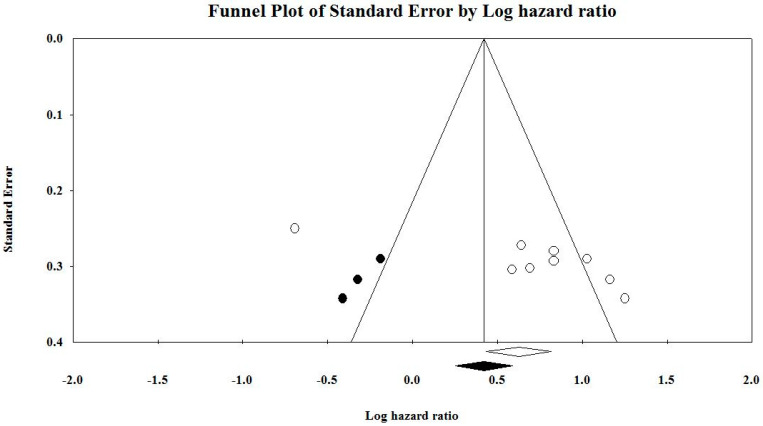
Funnel plot with imputed studies. The black circles represent imputed studies.

**Table 1 cancers-12-01199-t001:** Description of the 17 included studies.

Author	Country	Patients Origin	Ethnicity	No. of Samples (Cancer/Normal)	Cell Lines	Resistant Cells	miRNA	miRNA Profiling Platform	Drug	Pathways/Gene
Xu Bin et al., (2011) [30]	China	China	Chinese	9	DU145 and PC3	NM	143	Standard SYBR Green PCR kit (Roche) protocol on the 7300 real-time instruments.	Doc	V-Ki-ras2 Kirsten rat sarcoma viral oncogene homolog (KRAS)
Su S-F et al., (2012) [12]	USA	USA	Americans	7/7	Normal fibroblast LD419, non-tumorigenic human urothelial UROtsa and NK2464 cells	NM	30d, 181a and 199a-5p	miRNA Taqman assays (Applied Biosystems, Foster City, CA, USA)	Tg and TSA	Glucose-regulated protein-78 (GRP78)
Puhr M et al., (2012) [50]	Austria	Austria	Austrians	28	Human prostate cell lines PC3 and DU-145	PC3-DR and DU-145-DR	200c and 205	ABI Prism 7900HT system.	Doc	E-cadherin
Singh Saurabh et al., (2012) [11]	USA	USA	Americans	3	human metastatic prostate cancer cell lines DU145 and PC3 and their PTX resistant versions DU145-TXR and PC3-TXR	NM	10b, 17, 29b, 34a, 155 and 200c	SYBR Green dye universal master mix	PTX, CYCL	Hedgehog pathway
Yu Junjie et al., (2013) [31]	China	China	Chinese	30/15	Human PCa cell lines LNCaP, PC3 and DU145, BPH1	NM	200b	PrimeScript Reverse Transcription System and SYBR Premix Ex Taq™ II kit	Doc	B-cell-specific Moloney murine leukemia virus insertion site 1 (Bmi-1)
Zhang Q et al., (2014) [51]	USA	USA	Americans	8/8	WPE1-NA22, PC-3, and DU-145	PC3 and DU145	205,31	TaqMan assays Applied Biosystems (Foster City, CA, USA).	Doc and DZNep	Enhancer of zeste homolog 2 (EZH2)
Siu K M et al., (2014) [51]	China	China	Chinese	25	DU145/RasG37 cell line	NM	203	mirVana PARIS RNA isolation system	TKI	EGFR pathway
Lin H-M et al., (2014) [52]	Australia	Australia	Australians	97	PC3 and DU145 cell lines	PC3Rx and DU145Rx	20, 20a, 20b, 21, 25, 132, 146a, 200a, 200b, 200c, 222, 301b, 375, 429 and 590-5p	Taqman Array microRNA cards	Doc	NM
Liu F et al., (2015) [9]	China	China	Chinese	40	PC3 and LNCap	Cisplatin resistant PC cells	144	QuantiTect SYBR Green PCR Kit (Qiagen)	cDDP	VEGF
Yang Y et al., (2015) [53]	USA	USA	African Americans (AA) and Caucasian Americans (CA)	300	LNCap, MDAPCa-2b cells	hnRNPH1-expressing MDAPCa-2b cells	22, 132, 212 and 495	MessageAmp aRNA Kit	Bicalutamide	hnRNPH1
Liao H et al., (2016) [10]	China	NM	NM	25/25	Human PCa cell lines, including PC-3, DU145 and RWPE-1.	PC3 and DU145	34a	ABI 7500 thermocycler; Life Technologies (Bio-Rad, Hercules, CA, USA).	AZA, Topotecan and Dox	AMPK/mTOR pathway
Chen L et al., (2017) [8]	China	China	Chinese	74/28	PC3	PC3/PTX	199a	Eppendorf Mastercycler and Power SYBRâ„¢ Green Master Mix (ThermoFisher Scientific, USA)	PTX	Yamaguchi sarcoma viral homolog 1 (YES1)
Wu G et al., (2017) [54]	China	China	Chinese	124/30	PC3	Docetaxel Resistant PC-3 sub-lines (PC-3-R)	204	iCycler iQ™ Real-Time PCR Detection System (Bio-Rad Laboratories, Hercules, CA, USA).	Doc	Zinc-finger E-box-binding homeobox 1 (ZEB1)
Siu MK et al., (2017) [55]	China	NM	NM	40	DU145, PC3, LNCaP, RasB1 and 22Rv1	NM	1	TaqMan MicroRNA Assays kit (Applied Biosystems).	DHT and AR inhibitor (MDV3100)	Transcription Factor 7 (TCF7)
Liu J et al., (2017) [56]	China	NM	NM	44/30	prostate epithelial cells (PECs), LNCaP and PC-3 cells	Dox-resistant PC-3R cells	125a-3p	Tagman microRNA and mRNA assays.	Doc	Metastasis-associated protein 1(MTA1)
Lin X et al., (2018) [57]	China	China	Chinese	40	PC3 and HPrEC	NM	4319	Omniscript reverse transcription kit (Qiagen).	Estramustine	HER-2
Wang D et al., (2018) [58]	China	China	Chinese	25	PC-3 and LNCap	NM	182	miScript SYBR Green PCR Kit (Qiagen) on the DA7600 Real-time Nucleic Acid Amplification Fluorescence Detection System (Bio-Rad).	androgen-deprivation therapy	Wnt/ß-catenin signal pathway

Abbreviations: PTX—paclitaxel; Doc—docetaxel; DHT–dihydrotestosterone; cDDP—cisplatin; AR—androgen receptor; AZA—azacytidine; DZNep-3′ deazaneplanocin A; TKI–tyrosine kinase inhibitor; Tg—thapsigargin; TSA—trichostatin A; CYCL—cyclopamine; Dox—doxorubicin.

**Table 2 cancers-12-01199-t002:** Genetic pathways involved in chemoresistance.

Chemoresistance					
	Downregulated			Upregulated	
Drug	miRNA	Pathway	Drug	miRNA	Pathway
**Paclitaxel (PTX)**	199a	Yamaguchi sarcoma viral homolog 1 (YES1)	**Paclitaxel**	10b	Hedgehog pathway
**Docetaxel**	31	Enhancer of zeste homolog 2 (EZH2)		17	Hedgehog pathway
	125a-3p	Metastasis-associated protein 1(MTA1)		155	Hedgehog pathway
	143	V-Ki-ras2 Kirsten rat sarcoma viral oncogene homolog (KRAS)	**Cyclopamine**	10b	Hedgehog pathway
	204	Zinc-finger E-box-binding homeobox 1 (ZEB1)		17	Hedgehog pathway
	200b	B-cell-specific Moloney murine leukemia virus insertion site 1 (Bmi-1)		155	Hedgehog pathway
	200c	E-cadherin			
	205	Enhancer of zeste homolog 2 (EZH2)			
	205	E-cadherin			
**Dihydrotestosterone (DHT)**	1	AR signalling pathway, Transcription Factor 7 (TCF7)			
**AR inhibitor (MDV3100)**	1	AR signalling pathway, Transcription Factor 7 (TCF7)			
**Cisplatin**	144	VEGF			
**DZNep**	31	Enhancer of zeste homolog 2 (EZH2)			
	205				
**Tyrosine kinase inhibitor**	203	EGFR pathway			
**Paclitaxel**	29b	Hedgehog pathway			
	34a	Hedgehog pathway			
	200c	Hedgehog pathway			
**Cyclopamine**	29b	Hedgehog pathway			
	34a	Hedgehog pathway			
	200c	Hedgehog pathway			
**Trichostatin A**	30d				
	181a	Glucose-regulated protein-78 (GRP78)			
	199a-5p	Glucose-regulated protein - 78 (GRP78)			
**Thapsigargin**	30d				
	181a	Glucose-regulated protein-78 (GRP78)			
	199a-5p	Glucose-regulated protein-78 (GRP78)			

**Table 3 cancers-12-01199-t003:** Genetic pathways affecting chemosensitivity.

Chemosensitivity					
	Downregulated			Upregulated	
Drug	miRNA	Pathway	Drug	miRNA	Pathway
Azacytidine	34a	AMPK/mTOR pathway	-	-	-
Androgen receptor (AR) expression	212	hnRNPH1	-	-	-
Bicalutamide	212	hnRNPH1	-	-	-

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
