# Peer review of "Clinical Theragnostic Potential of Diverse miRNA Expressions in Prostate Cancer: A Systematic Review and Meta-Analysis"

_cancers, 2020, doi:10.3390/cancers12051199_

Round 1

Reviewer 1 Report

The purpose of this systematic review is to investigate and evaluate the miRNA biomarkers as potential predictors in chemoresistance/sensitivity in PrC, and their association with different drug regulatory genetic pathways.

In introduction, please describe how miRNA works as a potential predictor in chemoresistance/sensitivity in PrC.

The conclusion is too superficial. No insight.

You are doing a review on published results. You need to provide your own insight into this area.

Author Response

To

Ms. Treena Guo

Assistant Editor

treena.guo@mdpi.com

Cancers (MDPI)

Special issue: MicroRNA Therapeutics: Towards a New Era for the Management of Cancer

Dear Ms  Treena Guo,

We would like to thank once again the Cancers (MDPI), Editorial and Reviewer’s team for reviewing our manuscript and providing valuable comments. Our team firmly believe that the comments and feedback from your esteemed reviewers were constructive and would enhance the quality of the paper. Please find the revised manuscript to address issues identified by editors and reviewers team in considering the above review paper for publication.

We have amended the changes and highlighted the sections in the manuscript with track changes.

Reviewer 1

In introduction, please describe how miRNA works as a potential predictor in chemoresistance/sensitivity in PrC.

The authors’ thank the reviewer for their comments. We have added a paragraph about the role of miRNAs in diagnosis and prognosis in the introduction section (Line no: 121-124).

The conclusion is too superficial. No insight.

As per the reviewer suggests the conclusion has been modified (Line no: 406-416).

You are doing a review on published results. You need to provide your own insight into this area.

We have added an insight section in the conclusion part (Line no: 406-416).

Reviewer 2 Report

Since I am not an expert on either meta-analysis or systematic review, I cannot judge the statistical measures. However, their derived conclusion sounds biologically. It is worth while publishing in cancer.

Author Response

To

Ms. Treena Guo

Assistant Editor

treena.guo@mdpi.com

Cancers (MDPI)

Special issue: MicroRNA Therapeutics: Towards a New Era for the Management of Cancer

Dear Ms  Treena Guo,

We would like to thank once again the Cancers (MDPI), Editorial and Reviewer’s team for reviewing our manuscript and providing valuable comments. Our team firmly believe that the comments and feedback from your esteemed reviewers were constructive and would enhance the quality of the paper. Please find the revised manuscript to address issues identified by editors and reviewers team in considering the above review paper for publication.

We have amended the changes and highlighted the sections in the manuscript with track changes.

Reviewer 2

Since I am not an expert on either meta-analysis or systematic review, I cannot judge the statistical measures. However, their derived conclusion sounds biologically. It is worth while publishing in cancer.

The authors’ thank the reviewer for the comment.

Reviewer 3 Report

  1. Line 66: the remaining miRNA number should be 19, that equals 42-2-21
  2. Line 72,73: rephrase the sentence
  3. Line 74: keep the same format, Capitalization or not
  4. Introduction paragraph 1: describe the trend of PrC recent years, the number of death or the mortality rate. For paragraph 2, make it shorter and clear. For paragraph 3, add the description about chemosensitivity. 
  5. Table 1: re-organize the article in order by year.
  6. Line 289: figure 4 is not high-resolution and the format of legend needs to be changed. 
  7. Result 3.2.: use a figure to summarize cell line information.
  8. Line 413: pancreatic cancer patients should be prostate cancer patients.
  9. How about the miRNA expression level in docetaxel treated patient? since it is the most commonly used drug as mentioned.

Author Response

To

Ms. Treena Guo

Assistant Editor

treena.guo@mdpi.com

Cancers (MDPI)

Special issue: MicroRNA Therapeutics: Towards a New Era for the Management of Cancer

Dear Ms  Treena Guo,

We would like to thank once again the Cancers (MDPI), Editorial and Reviewer’s team for reviewing our manuscript and providing valuable comments. Our team firmly believe that the comments and feedback from your esteemed reviewers were constructive and would enhance the quality of the paper. Please find the revised manuscript to address issues identified by editors and reviewers team in considering the above review paper for publication.

We have amended the changes and highlighted the sections in the manuscript with track changes.

Reviewer 3

The authors’ thank the reviewer for their comments and suggestion.

  1. Line 66: the remaining miRNA number should be 19, that equals 42-2-21

We have modified the respective statement to correct the factualness (Line no:66).

  1. Line 72,73: rephrase the sentence

We have rephrased the sentence from line no: 71-73.

  1. Line 74: keep the same format, Capitalization or not

We have modified the keywords (Line no: 74).

  1. Introduction paragraph 1: describe the trend of PrC recent years, the number of death or the mortality rate. For paragraph 2, make it shorter and clear. For paragraph 3, add the description of chemosensitivity. 

We have modified the 2nd paragraph and have mentioned a line about the chemosensitivity in line no: 83-97. A sentence about chemosensitivity is added on line no: 105.

  1. Table 1: re-organize the article in order by year.

As per the reviewer suggestion, we have re-ordered the table based on the year (Line no: 214-216).

  1. Line 289: figure 4 is not high-resolution and the format of legend needs to be changed. 

As per the reviewer’s suggestion, we have improved the quality of  Figure 4 and changed the format of the legends (line no: 288-291).

  1. Result 3.2.: use a figure to summarize cell line information.

As per the reviewer suggestion, a panel has been added to Figure 2B to summarise the cell line information (line no: 227)

  1. Line 413: pancreatic cancer patientsshould be prostate cancer patients.

We have modified changed the word from “pancreatic” to “prostate” (Line no: 410).

  1. How about the miRNA expression level in docetaxel treated patient? Since it is the most commonly used drug as mentioned.

A total of 20 different miRNAs have been studied under the docetaxel treatment while 2 have been found to be upregulated (132 and 301b) while 18 have been downregulated (20a, 20b, 21, 25, 31, 125a-3p, 143, 146a, 200a, 200b, 200c, 204, 205, 205, 222, 375, 429 and 590-5p).

Reviewer 4 Report

Brief Summary: The aim of the study by Jayaraj et al., was to assess the impact of microRNA (miRNA) expression on prostate cancer (PCa) chemo-resistance and -sensitivity. The authors performed a systematic review and meta-analysis of published data and report several miRNAs to be associated with PCa chemotherapy responses, and touch upon the potential contributions of these miRNAs to PCa biology and cancer management. They incorporate both immortalized cell line data as well as clinical human PCa data to identify miRNAs of interest. Several studies have implicated miRNAs to cellular processes regulating treatment response in PCa, especially in androgen-response settings, and the present data can serve as a cumulation of these studies and potentially a valuable resource for investigators interested to further study the prognostic and therapeutic potential of miRNAs.

Strengths of the study:

  • Original systematic literature review pipeline following established (PRISMA guidelines) selection criteria to discover targets of interest.
  • Valuable resource for assessing the prognostic role of miRNAs in PCa patients.

Weaknesses of the study:

  • The scope of the study span only until end of 2018. However, several impactful original as well as review studies have been published in 2019 that may hinder the novelty or/and importance of the present manuscript. Examples include but not limited to, Khorasani et al., 2019 PMID: 32215262; Guan et al., 2019 PMID: 31949655 and Fu et al., 2019 PMID: 31889959.
  • Publication bias may have hindered the inclusion of important studies that could explain the under-representation of androgen-dependent chemotherapy responses.
  • Lack of data for PCa patients of African-origin, known to have the most aggressive form of the disease (Lack of studies performed with this patient population? Authors can clarify).

Specific Comments: The authors demonstrate that miRNAs facilitate chemotherapy drug resistance and sensitivity in PCa. However, there are several issues to be addressed in their study.

Abstract: Nice summary of the study outlining the objective, major findings and methods used as well as conclusion and significance.

Introduction: This section is well structured and balanced in terms of background information on PCa, PCa treatment and chemoresistance as well as the important role of mRNAs in PCa. The authors highlight the gap in literature and the significance of their study. Some comments:

  • More recent epidemiological studies have been performed and the authors should update or remove their PCa-related death statistics from 2012 [minor].
  • Androgen depravation therapy is standard of treatment for PCa. Taxane chemotherapy comes as a second or third line of therapy, as the authors also point out, in Androgen-Independent PCa (AIPCa). Would the study benefit more and be more clinically relevant If the focus of the human data meta-analysis was AIPCs? [major]

Materials and Methods: The experimental procedures and study design are adequately explained, and details are properly provided.

Results: Overall results and figures are presented in orderly and comprehensible fashion. The authors describe the stud designs and data analysis, including the miRNA-mediated drug regulatory pathways in chemotherapy responses as well as the association of miRNA expression to patient survival. Comments:

  • The data presented in Table 1 is greatly appreciated as it nicely summarizes the studies selected for analyses and their main characteristics.
  • Section 3.2 and Figure 2 detail the in vitro assays used to assess the expression and functional role of miRNAs in the selected studies. Therefore, the section title should be modified to avoid confusion [minor]. Moreover, the authors mention in vivo assays but do not show any specific in vivo assays. What do they define as in vivo? Xenograft tumor models? [minor]
  • Sections 3.4 belongs to the Materials and Methods [minor].
  • Figure 4 requires quality improvement since it is not readily comprehensible [minor].
  • Sections 3.7.2, 3.7.3, 3.7.4 and 3.7.5 should be merged and summarized [minor].
  • As both the 3.7.1 and 3.7.6 sections deal with the bias introduced in the study, they should also probably be merged [minor].
  • It is unclear what the difference between Figure 5 and 6 is. In the first case (Figure 5) the funnel plot suggested a publication bias in the study whereas, in the second case (Figure 6, with imputed studies) it does not. What do the authors means by ‘impute studies’? They should clarify this discrepancy and simplify the wording [major].

Discussion and Conclusions: The authors nicely discuss their findings and put them in context of published data. They also briefly discuss the strengths and limitations of their analyses. However, this section needs improvement and more careful proof-reading. Comments:

  • The authors mention that a previous study implicated miRNA-21 in PCa androgen response and treatment. Although miRNA-21 was shown to be down-regulated in their PCa analyses, it did not significantly associate with chemo-resistance or -sensitivity. The authors could discuss the potential reasons for that [minor].
  • In the 5th paragraph of this section, the role of various proteins in PCa biology and therapy is outlined and discussed. Although the authors summarize the targets of the identified miRNAs in the Results sections and Table 2, they should specifically include them for each of the targets mentioned here (e.g. miRNA-31 and mRNA-205 target EZH2). This will put their findings into more context in text [major].
  • The study would benefit if the authors could provide an explanation as to why their study included only Asian and Caucasian PCa patient populations and not any of African origin. Was this a result of the analyses criteria? [major]
  • Lack of proof-reading. In parts of the Discussion and Supplementary Table 1, the authors refer to T-cell Acute Lymphoblastic leukemia (T-ALL) or pancreatic cancer, instead of PCa [major].

Author Response

To

Ms. Treena Guo

Assistant Editor

treena.guo@mdpi.com

Cancers (MDPI)

Special issue: MicroRNA Therapeutics: Towards a New Era for the Management of Cancer

Dear Ms  Treena Guo,

We would like to thank once again the Cancers (MDPI), Editorial and Reviewer’s team for reviewing our manuscript and providing valuable comments. Our team firmly believe that the comments and feedback from your esteemed reviewers were constructive and would enhance the quality of the paper. Please find the revised manuscript to address issues identified by editors and reviewers team in considering the above review paper for publication.

We have amended the changes and highlighted the sections in the manuscript with track changes.

Reviewer 4

The authors’ thank the reviewer for their critical evaluation and suggestion of our manuscript and we have incorporated the changes to improve the scientific and technical quality.

Weaknesses of the study:

  • The scope of the study span only until end of 2018. However, several impactful original as well as review studies have been published in 2019 that may hinder the novelty or/and importance of the present manuscript. Examples include but not limited to, Khorasani et al., 2019 PMID: 32215262; Guan et al., 2019 PMID: 31949655 and Fu et al., 2019 PMID: 31889959.

As per the reviewer’s suggestion we have searched the literature and found eight studies to have been obtained while none of the studies complied with our selection criteria. Of the 8, four were reviews, three studies were performed only in cell lines and one belonged to another cancer type. The articles which reviewers suggested belong to our exclusion criteria where few were reviews and one had only data on cell lines.

  • Publication bias may have hindered the inclusion of important studies that could explain the under-representation of androgen-dependent chemotherapy responses.
  • Lack of data for PCa patients of African-origin, known to have the most aggressive form of the disease (Lack of studies performed with this patient population? Authors can clarify).

As we have mentioned in our selection criteria, the articles were not eliminated based on any particular geographical location, as the reviewer has mentioned it might be due to the lack of studies with this patient population.

Some comments:

  • More recent epidemiological studies have been performed and the authors should update or remove their PCa-related death statistics from 2012 [minor].

We have updated the epidemiological data in the introduction section with the 2018 GLOBOCAN statistic (Line no: 78-81).

  • Androgen depravation therapy is standard of treatment for PCa. Taxane chemotherapy comes as a second or third line of therapy, as the authors also point out, in Androgen-Independent PCa (AIPCa). Would the study benefit more and be more clinically relevant If the focus of the human data meta-analysis was AIPCs? [major]

Our focus was not on any specific treatment methodology due to which we have included all forms of treatment in the review, due to the lack of studies we were not able to perform sub-group analysis which we have done for our other cancer papers based on the types of miRNA’s, treatment, demographic characteristics, etc. (https://www.mdpi.com/2072-6694/11/7/900/htm, https://link.springer.com/article/10.1007/s40291-019-00440-y and https://www.ncbi.nlm.nih.gov/pmc/articles/PMC7008181/)

  • Section 3.2 and Figure 2 detail the in vitro assays used to assess the expression and functional role of miRNAs in the selected studies. Therefore, the section title should be modified to avoid confusion [minor]. Moreover, the authors mention in vivo assays but do not show any specific in vivo assays. What do they define as in vivo? Xenograft tumor models? [minor]

As per the reviewer suggestion the title of the section 3.2 is renamed as “In vitro assays” (line no: 218). We have collated the information regarding the diverse experimental evidences with the in vivo assays from the included articles and have listed out the commonly performed ones to demonstrate the way in which miRNAs have been validated. Xenograft tumour models and mice model studies are included under the in vivo category.

  • Sections 3.4 belongs to the Materials and Methods [minor].

We would like to point out that the data provided in section 3.4 is the number of studies involving different treatment strategies and it was extracted from the included articles, technically these are our part of the results.

  • Figure 4 requires quality improvement since it is not readily comprehensible [minor].

As per the reviewer’s suggestion, we have improved the quality of the Figure 4.

  • Sections 3.7.2, 3.7.3, 3.7.4 and 3.7.5 should be merged and summarized [minor].

The sections separated from 3.7.2-3.7.6 is because these are different publication bias and collating them together will lead to confusion as differed tests mean different inference.

  • As both the 3.7.1 and 3.7.6 sections deal with the bias introduced in the study, they should also probably be merged [minor].

Similar to our previous response the section 3.7.1 represents the funnel plot wherein we calculate the bias based on the included studies while 3.7.6 discusses the bias about the missing studies calculated using Duval and Tweedie's Trim and Fill bias indicator test.

  • It is unclear what the difference between Figure 5 and 6 is. In the first case (Figure 5) the funnel plot suggested a publication bias in the study whereas, in the second case (Figure 6, with imputed studies) it does not. What do the authors means by ‘impute studies’? They should clarify this discrepancy and simplify the wording [major].

The imputed studies mean that the addition of missing studies due to the publication bias which are calculated based on the Duval and Tweedie’s trim and fill method where in it calculates the number of left out studies (Dark black dots) and by including them what could be the effect of the publication bias. As per our study, there are three studies left out and by including them it leads to asymmetry as it falls outside the funnel plot.

Discussion and Conclusions: The authors nicely discuss their findings and put them in the context of published data. They also briefly discuss the strengths and limitations of their analyses. However, this section needs improvement and more careful proof-reading. Comments:

  • The authors mention that a previous study implicated miRNA-21 in PCa androgen response and treatment. Although miRNA-21 was shown to be down-regulated in their PCa analyses, it did not significantly associate with chemo-resistance or -sensitivity. The authors could discuss the potential reasons for that [minor].

We have briefly discussed regarding the down-regulation similar to the previous reports (Line no: 343-346).

  • In the 5thparagraph of this section, the role of various proteins in PCa biology and therapy is outlined and discussed. Although the authors summarize the targets of the identified miRNAs in the Results sections and Table 2, they should specifically include them for each of the targets mentioned here (e.g. miRNA-31 and mRNA-205 target EZH2). This will put their findings into more context in text [major].

We have obtained a total of 42 miRNAs and 14 different drug regulated pathways it is difficult to elaborate every pathway with its target miRNA, due to which we have represented them in the table as well as classified the gene’s/pathways based on the cellular role such as cell survival, cell differentiation, proliferation and angiogenesis and have represented as Figure 3.

  • The study would benefit if the authors could provide an explanation as to why their study included only Asian and Caucasian PCa patient populations and not any of African origin. Was this a result of the analyses criteria? [major]

We have not limited our searches on any country of origin, race or any other populations, the results were analysed for the availability of the selection criteria. The reason to choose both patients as well as in vitro data was that the HR value could be used for the meta-analysis while the cell line data and the regulation and treatment exposures of miRNA can be used for the pathway analysis. We would like to point out that while screening for the articles with the respective data we found that 34 studies had only data on cell lines while 3 were only with patients due to which we could not include them in our systematic review and meta-analysis.

  • Lack of proof-reading. In parts of the Discussion and Supplementary Table 1, the authors refer to T-cell Acute Lymphoblastic leukemia (T-ALL) or pancreatic cancer, instead of PCa [major].

The authors are thankful for the reviewers in pointing out these errors we have revised them throughout the draft.

Round 2

Reviewer 1 Report

Check English .

Reviewer 3 Report

The authors Jayaraj, et. al. have systematically summarized the study of miRNA expressions and their role in chemoresistance in prostate cancer from 2909 publications retrieved. The work is labor intensive and very meaningful, it depicts the significance of miRNA expression as theragnostic biomarkers in medical oncology. After the major revision, the content and the figures in the manuscript are all much better, which gives readers a clear view. 

Reviewer 4 Report

The authors have addressed the majority of reviewer’s comments and the manuscript has improved. Although an in-text response to some of the comments raised could have been included in the manuscript text, the authors offer clear explanations to reviewer’s concern in their responses.